# A Novel QoS Guaranteed Joint Resource Allocation Framework for 5G NR with Supplementary Uplink Transmission

Yanzan Sun, Yanyu Huang, Tao Yu, Xiaojing Chen and Shunqing Zhang *

School of Communication and Information Engineering, Shanghai University, 99 Shangda Road, Shanghai 200444, China
* Correspondence: shunqing@shu.edu.cn

**Abstract:** In 5G scenarios, the dynamic resource allocation of network slicing is crucial for quality-of-service (QoS) guaranteed under fluctuating traffic demands in rapidly changing communication environments. In this paper, we propose a novel QoS guaranteed joint resource allocation framework for NR with supplementary uplink (SUL) called QGJRA-SUL, where three parameters of SUL admission, TDD pattern, and band slicing scheme are jointly optimized. The framework is driven by a well-designed deep reinforcement learning agent. By combining the activation functions tanh and softmax, the agent can jointly optimize three parameters at the same time. Under the original problem of QoS satisfaction rate maximization, we introduce the load unbalance degree of slices into the reward function as a penalty term. The simulation results show that the framework can guarantee the QoS satisfaction rate well and balance the load of slices. QGJRA-SUL can accommodate 15% more user equipments (UEs) with the same QoS satisfaction rate than that of a traditional single-band solution without SUL, and achieve a 73% increase in the performance of load balancing than that without a load balancing mechanism near the full load.

**Keywords:** network slicing; supplementary uplink; deep reinforcement learning

## 1. Introduction

With the emergence of novel smart applications such as smart homes, online education, HD live streaming, etc., both downlink and uplink traffic demands fluctuate dramatically [1]. Furthermore, the utilization of downlink and uplink is affected by the greater path loss of the NR band with higher frequency [2] and a low uplink to downlink duration ratio as a common used configuration in the time-division-duplex (TDD) system [3]. Especially, the uplink channel quality of cell edge user equipments (UEs) will deteriorate severely due to the inter-cell interference and a low uplink transmission power, resulting in the uplink coverage being shrinked compared to the downlink coverage. To better meet the diverse quality-of-service (QoS) of novel services, 5G NR (5th Generation New Radio) enables several promising features for more flexible radio resource allocation, such as network slicing [4], dynamic TDD, Supplementary Uplink (SUL) [5], etc. The combination of various features needs to be studied to further improve the QoS satisfaction rate.

Firstly, network slicing is a basic service form oriented to vertical industries in the 5G era, enabling the flexible allocation of network resources. For example, a flexible soft spectrum slicing based on a predictive approach was studied under asymmetric traffic conditions [6], and a dynamic radio resource slicing framework was proposed for QoS, guaranteed by the joint optimization of band slicing ratio and UE association for a two-tier heterogeneous wireless network [7]. A LSTM-A2C architecture is proposed to track user mobility and to improve the utility of network slice resource allocation [8]. Another LSTM-DQN architecture is proposed to improve the performance during the initial exploration phase [9]. Secondly, dynamic TDD can provide more flexibility to accommodate the downlink and uplink traffic demands between different cells, and then improve the QoS of

downlink and uplink [10]. For instance, a dynamic TDD frame structure adaptation scheme was proposed to control the cross-link interference among cells [11]. Deep reinforcement learning (DRL)-based TDD pattern adaptation schemes were proposed according to the downlink and uplink buffer status, as well as the mobility behaviors [12,13]. Thirdly, 3rd Generation Partnership Project (3GPP) R15 introduced the SUL in the Sub-3G [5,14], which is a different method of spectrum configuration compared with NR UL. 5G NR introduces downlink and uplink decoupling technology, which can transfer the cell edge UEs' uplink transmission to the SUL band. Moreover, SUL is the term for both the frequency division duplexing (FDD) UL band and an uplink enhancement technology with many advantages: a larger coverage area, more uplink spectrum resources, a lower delay, and lower deployment costs [3,15,16], and as a newly defined frequency band, it can also make the existing dual connectivity and carrier aggregation achieve higher performances. Ref. [17] discussed two deployment modes, which are independent SUL deployment and SUL and NR FDD co-deployment. An LTE load-based downlink and uplink decoupling mode are discussed in [18]. Furthermore, Ref. [19] discussed the UEs' correlation probability, and impacts on throughput in two-layer heterogeneous networks were discussed.

To sum up, all of the above three methods can meet UEs' QoS and throughput requirements by means of resource allocation, and then inevitably interact with each other. However, the current research rarely jointly considers these three methods' effects. It is necessary to develop intelligent and dynamic resource allocation to maximize the QoS satisfaction rate, especially with network slicing and SUL configuration [20]. For this purpose, we propose a novel QoS guaranteed joint resource allocation framework for NR with SUL (QGJRA-SUL), which enables a joint optimization of the dynamic SUL admission strategy and the dynamic TDD pattern, as well as the band slicing scheme in terms of bandwidth. The main contributions of this paper are listed below.

- Joint Optimization of SUL Admission, TDD Pattern, and Band Slicing Scheme: In the practical implementation of hard radio access network (RAN) slicing, each BS needs to switch the downlink/uplink transmission of all the RAN slices simultaneously, according to different TDD patterns [21]. With SUL-enabled transmission, the admission results of the SUL band will directly affect the TDD pattern and the band slicing scheme. To solve this issue, we formulate a joint optimization problem of SUL admission, TDD pattern, and band slicing scheme, and propose a well-designed DRL-based solution. The activation function of the output layer is well-designed to accommodate the three optimization aspects mentioned above.
- Load Balance of Slices for further QoS Guarantee:Traditional load balancing mainly focuses on different nodes of the core network or multiple RANs [22–24]. Rapidly fluctuating traffic requires a solution with a margin for each slice-level service enabled with SUL. We introduce the slice-level load balancing mechanism into the above DRL-based solution for further QoS guaranteed in the face of fluctuating traffic demands.

The rest of this paper is organized as follows. Section 2 introduces the system model and the corresponding wireless resource allocation. In Section 3, we formulate an optimization problem regarding QoS satisfaction rate maximization, as well as load balancing. In Section 4, we transform the original optimization problem into a Markov Decision Process (MDP) and propose a DRL-based solution of the Deep Deterministic Policy Gradient (DDPG) algorithm. In Section 5, we compare the proposed framework with the baselines and analyze the performance, followed by the concluding remarks in Section 6.

## 2. System Model

In this section, we provide the mathematical model of the TDD network, which is enabled with network slicing and SUL. The corresponding performance metrics are defined as well.

### 2.1. Communication Model

As depicted in Figure 1, we consider a 5G TDD system that is enabled with network slicing and SUL transmission. The SUL band has a larger coverage than the NR band [5]. Network slicing enables the BS to provide services with different QoS requirements, and it maintains isolation between services. The whole bandwidth is divided into two parts, i.e., NR band with bandwidth $W^{nr}$ for both the downlink and uplink transmission in the TDD pattern, and the SUL band with bandwidth $W^{sul}$ only for SUL transmission. UEs are distributed in the coverage of the BS randomly. Let $\mathcal{M} = \{1, 2, \cdots, m, \cdots, M\}$ denote the slice set, and $\mathcal{N}_m$ denote the set of UEs belonging to slice $m$. It is assumed that each UE belongs to only one slice, that is, one UE uses only one service. UEs in different slices with different QoS metrics are marked with different colors in Figure 1. This paper focuses on a scenario where each slice has different downlink and uplink rate requirements. Key notations that are used in this paper are presented in Table 1. To reasonably allocate the resources of the SUL band, we first define a SUL admission identifier $x$, based on the reference signal received power(RSRP), as below:

$$x = \begin{cases} nr & RSRP \geq \delta \\ sul & RSRP < \delta \end{cases} \tag{1}$$

where $\delta$ is the RSRP threshold to decide the SUL band admission. In other words, UEs with $x = sul$ are classified as edge UEs of the cell, using the SUL band for uplink transmission. Obviously, the downlink communication can only use the NR band for transmission. Note that UE may be scheduled to transmit either on the SUL or on NR-UL, but not on both at the same time [5]. Furthermore, the switching time between the NR UL carrier and the SUL carrier is 0 is [14].

**Table 1.** Description of main notations.

| Notation | Description |
|---|---|
| $W^{nr}$ | The bandwidth of the NR band. |
| $W^{sul}$ | The bandwidth of the SUL band. |
| $x$ | The identifier for SUL admission. |
| $\mathcal{M}$ | The set of slices provided by the BS. |
| $\mathcal{N}_m$ | The set of UE belonging to slice $m$. |
| $\gamma_{m,n}^d$ | The SNR of UE $n$ of slice $m$ in direction $d$. |
| $r_{m,n}^d$ | The achievable rate of UE $n$ under $\gamma_{m,n}^d$ on one RB. |
| $R_{m,n}^d$ | The achievable rate of UE $n$ in the period $T$. |
| $\delta$ | The RSRP threshold to decide SUL band admission |
| $\lambda$ | TDD pattern |
| $\tau$ | The duration of a TDD pattern |
| $T$ | The period where $\lambda$ remains the same |
| $w_m^d$ | The bandwidth allocated to slice $m$ in direction $d$. |
| $\eta_m^{d,nr}$ | The load of slice $m$ in direction $d$ of NR band |
| $\eta^{sul}$ | The load of SUL band |
| $q_{m,n}^d$ | The UE-level QoS satisfaction rate of UE $n$ of slice $m$ in direction $d$. |
| $q_m^d$ | The slice-level QoS satisfaction rate of slice $m$ in direction $d$. |
| $q$ | The BS-level QoS satisfaction rate. |
| $\varsigma$ | The variance of slice load |
| $\alpha_m^d$ | The weight of slice $m$ in direction $d$ |
| $\beta$ | The weight of $\varsigma$ to make the trade-off with $q$ |
| $\bar{R}_m^d$ | The throughput requirement of slice $m$ in direction $d$. |

We denote $P_{m,n}^{dl,x}$ and $P_{m,n}^{ul,x}$ to represent the transmission power in downlink and uplink, respectively. $H_{m,n}^{dl,x}$ and $H_{m,n}^{ul,x}$ represent the downlink and uplink path loss between UE $n$ of slice $m$ and the BS, respectively. For the sake of simplicity, the communication direction set is denoted as $\mathcal{D} = \{d | d = dl \ or \ d = ul\}$, where $d = dl$ and $d = ul$ represent the downlink

and uplink, respectively. For UE $n$ belonging to slice $m$, its SNR in direction $d$ can be expressed as

$$\gamma_{m,n}^{d,x} = \frac{P_{m,n}^{d,x} H_{m,n}^{d,x}}{N_0} \tag{2}$$

Note that $x$ is always $nr$ for downlink transmission. $N_0$ is the power of additive white Gaussian noise. Assume that each resource block (RB) has a bandwidth $b$, then the achievable rate of UE $n$ in direction $d$ on one RB can be expressed as

$$r_{m,n}^{d,x} \leqslant b \times \log_2\left(1 + \gamma_{m,n}^{d,x}\right). \tag{3}$$

The rate that the UE $n$ of slice $m$ in direction $d$ can achieve in each period of resource-slicing is denoted as $R_{m,n}^d$.

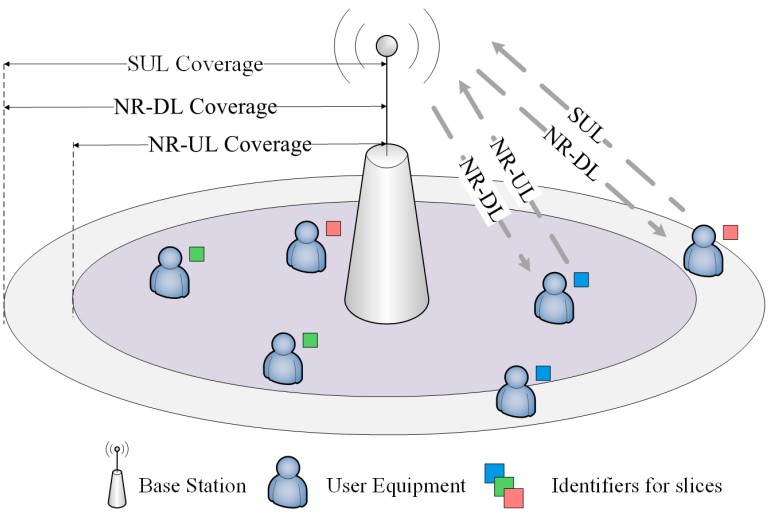

**Figure 1.** System Model. A 5G TDD BS is enabled with network slicing and SUL transmission. The edge UEs of the coverage can use the SUL band for uplink transmission. UEs in different slices are marked with different colors.

*2.2. Wireless Resource Slicing*

The 3GPP standard specifies 56 flexible slot formats [25]. For a scenario where the downlink traffic demand is greater than the uplink traffic demand, the BS can operate a configuration of slot format that assigns more RB and Orthogonal Frequency Division Multiplexing (OFDM) symbols for downlink transmission. In this paper, we consider a mode where each TDD pattern has only one downlink-to-uplink switch point. To align with the actual deployment, the entire spectrum of the NR band uses the same TDD pattern, that is, all slices switch synchronously between downlink and uplink. Therefore, the downlink and uplink bandwidths are both $W^{nr}$ in the frequency domain. The TDD pattern can be depicted by parameter $\lambda$, which represents the ratio of downlink duration to a TDD pattern duration. $\lambda$ is constrained by

$$0 < \lambda < 1. \tag{4}$$

Let $\tau$ denote the duration of a TDD pattern, which contains $N^{dl} + N^{ul}$ OFDM symbols. $N^{dl}$ and $N^{ul}$ denote the number of OFDM symbols for downlink and uplink, respectively. Here, we simplified the structures of pilots and special subframe, including ignoring DwPTS (Downlink Pilot Time Slot), GP (Guard Period), and UpPTS (Uplink Pilot Time Slot), which have no significant effect on the results of the experiment. Based on (4), $\lambda$ can be specifically defined as $\lambda = \frac{N^{dl}}{N^{dl} + N^{ul}}$. The TDD pattern $\lambda$ divides $\tau$ into downlink duration $\lambda\tau$ and uplink duration $(1 - \lambda)\tau$. The BS adjusts the allocation of downlink and uplink duration through $\lambda$, depending on the fluctuating slice load. For example, with the increasing need

for downlink, $\lambda$ becomes larger, which means that more resources are allocated to downlink. On the contrary, with the increasing need for uplink, $\lambda$ becomes smaller, which means that more resources are allocated to uplink. As shown in Figure 2, every time a period of time $T$ passes, $\lambda$ can be dynamically reconfigured to adapt to the new traffic ratio.

For the NR band, each slice is assigned a mutually orthogonal bandwidth for full isolation, i.e., hard slicing. Furthermore, the SUL band can be used by all slices as a common band, which is called soft slicing. The resource allocation diagram is shown in Figure 2. Let $w_m^d$ denote the bandwidth of slice $m$ assigned in the NR band. The hard slicing scheme can be expressed as a tuple $\mathbf{w} = \{w_m^d | w_m^d > 0, \sum_{m \in \mathcal{M}} w_m^d = W^{nr}, m \in \mathcal{M}, d \in \mathcal{D}\}$. At the beginning of the period of resource-slicing, the BS decides the SUL admission for user association, determines the TDD pattern for the whole NR spectrum, and allocates the NR bandwidth to all slices.

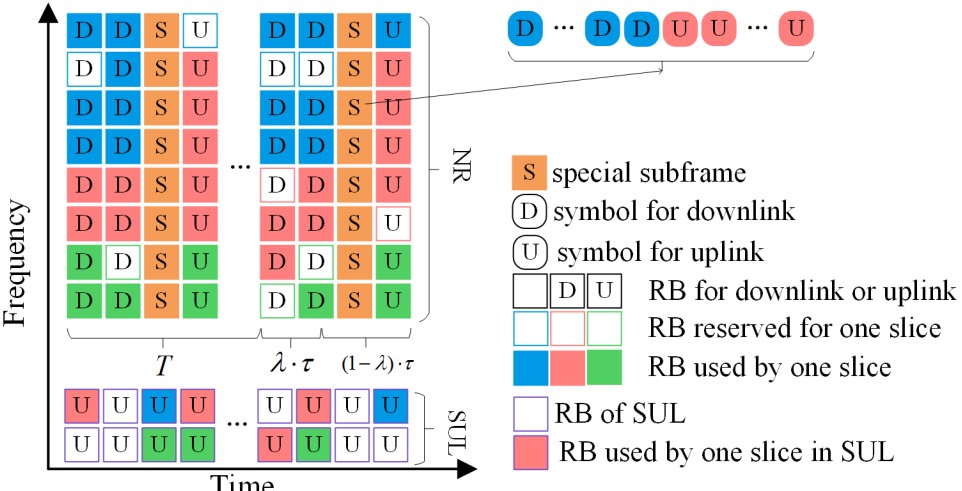

**Figure 2.** The joint resource allocation of NR band and SUL band. Each TDD pattern has only one downlink-to-uplink switch point. Hard slicing is operated in the NR band, and soft slicing is operated in the SUL band.

*2.3. Performance Metrics*

We take the throughput $R_{m,n}^d$ achieved by UE $n$ of slice $m$ in direction $d$ in the period $T$ as the object of QoS evaluation. Then, the QoS satisfaction rate $q_{m,n}^d$ of a UE $n$ can be expressed as

$$q_{m,n}^d = min(1, \frac{R_{m,n}^d}{\bar{R}_m^d}), \forall m \in \mathcal{M}, d \in \mathcal{D}, n \in \mathcal{N}_m, \tag{5}$$

where $\bar{R}_m^d$ is the QoS requirement of UE of slice $m$ in direction $d$. Based on the UE-level QoS satisfaction rate $q_{m,n}^d$, the slice-level QoS satisfaction rate of slice $m$ is expressed as

$$q_m^d = \frac{1}{|\mathcal{N}_m|} \sum_{n=1}^{|\mathcal{N}_m|} q_{m,n}^d. \tag{6}$$

Based on the slice-level QoS satisfaction rate $q_m^d$, the QoS satisfaction rate at the BS-level is defined as the normalized multi-slice QoS satisfaction rate, which is expressed as

$$q = \sum_{m \in \mathcal{M}} \sum_{d \in \mathcal{D}} \alpha_m^d \cdot q_m^d \tag{7}$$

where $\alpha_m^d$ is the weight of slice $m$ in direction $d$. The larger $\alpha_m^d$ is, the more important the corresponding slice $m$ in direction $d$ is. $\alpha_m^d$ is constrained by

$$\sum_{m \in \mathcal{M}} \sum_{d \in \mathcal{D}} \alpha_m^d = 1. \tag{8}$$

Let $\eta_m^{d,nr}$ and $\eta^{sul}$ denote the slice loads using the NR and SUL bands, respectively. The load-balancing metric of slices is defined as

$$\varsigma = var(\{ \bigcup_{m \in \mathcal{M}, d \in \mathcal{D}} \eta_m^{d,nr} \} \cup \{ \eta^{sul} \}), \tag{9}$$

where the function $var(\cdot)$ is to calculate the variance of slice load. The greater the load difference between slices, the greater the value of $\varsigma$.

### 3. Problem Formulation

In this section, we first formulate an optimization problem for our QoS-guaranteed joint resource allocation framework for NR with SUL. The proposed QGJRA-SUL framework involves three parameters for joint optimization, as described below:

1.  SUL admission: Considering the dynamic change of the wireless environment, SUL admission needs to be dynamically optimized by adjusting $\delta$.
2.  TDD pattern: As shown in Figure 2, a hard slicing scheme of the NR band for multiple slices with the same $\lambda$ is proposed. $\lambda$ needs to be dynamically optimized for the load between the downlink and the uplink.
3.  Band slicing scheme: The hard slicing scheme **w** represents an isolated bandwidth allocation scheme for both the downlink and uplink.

The detailed steps of the QGJRA-SUL framework are shown in Algorithm 1, where we assume that the BS continuously reconfigures the above three parameters $\delta, \lambda, \textbf{w}$ with period T. The description of the system state as the input of the DRL solution is detailed in Section 4.

---

**Algorithm 1** The QGJRA-SUL framework

---

1: **repeat**
2:      $\delta, \lambda, \textbf{w} \leftarrow$ DRL-based solution
3:      **for** slice $m = 1 \rightarrow M$ **do**
4:          **for** UE $n = 1 \rightarrow |\mathcal{N}_m|$ **do**
5:              **if** $\gamma_{m,i}^{dl,x} < \delta$ **then**
6:                  $x_{m,i} = sul$;
7:              **else**
8:                  $x_{m,i} = nr$;
9:              **end if**
10:          **end for**
11:      **end for**
12:      The TDD pattern $\lambda$ in Equation (4) divides $\tau$ into downlink duration $\lambda\tau$ and uplink duration $(1 - \lambda)\tau$ in the time domain.
13:      **for** slice $m = 1 \rightarrow M$ **do**
14:          **for** $d \in \mathcal{D}$ **do**
15:              Set the bandwidth allocated to slice $m$ to $w_m^d$;
16:          **end for**
17:      **end for**
18:      Base station provides communication services to UEs;
19:      **for** $t = 0 \rightarrow T - 1$ **do**
20:      **end for**
21:      System State $\rightarrow$ DRL-based solution;
22: **until** The Base station shutdown.

---

The objective of QGJRA-SUL is to guarantee the slice QoS satisfaction rate as much as possible while balancing the slice load via the joint optimization of the SUL admission, TDD pattern, and band slicing scheme. The utility function of each period $T$ can be formulated as

$$U = q - \beta \cdot \varsigma. \tag{10}$$

The first part of (10) is to maximize the QoS satisfaction rate, and the second part, $-\beta \cdot \varsigma$ is a penalty item to balance the slice load. $\beta$ is the weight of the penalty item to make the trade-off between the QoS satisfaction rate and the slice load balance. The optimal state of the system is that $q = 1$ and $\varsigma = 0$, which means the QoS satisfaction rate of each slice is 1, and all slices have the same load. With a traffic model $\mathbf{g} = \{g_{m,n}^d(t) | m \in \mathcal{M}, n \in \mathcal{N}, d \in \mathcal{D}, t \in [0, T-1]\}$, the QoS satisfaction rate $q(\lambda, \mathbf{w}, \delta, \mathbf{g})$ is influenced by the three parameters mentioned above and the traffic model. Hence, the problem is formulated as follows.

$$\mathbf{P} : \max_{\delta, \lambda, \mathbf{w}} q - \beta \cdot \varsigma$$

$$\text{s. t. } (1), (3), (4), (8),$$

$$0 < w_m^d, \tag{11}$$

$$\sum_{m \in \mathcal{M}} w_m^d = W^{nr} \tag{12}$$

## 4. Problem Transformation and DRL-Based Solution

The key challenge for solving the problem $\mathbf{P}$ is that $q(\lambda, \mathbf{w}, \delta, \mathbf{g})$ cannot be expressed by a closed-form mathematical model [26]. The difficulties in solving the problem $\mathbf{P}$ are reflected in the following aspects.

- *Diverse traffic models and QoS requirements:* Different slices have different traffic models and different QoS requirements. The resource allocation policy of a single slice cannot be directly applied to other slices.
- *Highly dynamic environment with user mobility:* Due to the varying network environment caused by the mobility of UEs and the uncertainty of task arrival, it is difficult to use concrete mathematical methods to allocate resources quickly.
- *Markovian characteristics of resource slicing:* The inter-slice resource allocation exhibits Markovian characteristics, where problems whose probability of the problem entering the next state depends only on the current state and action selected. In this paper, the allocation policy affects not only the current QoS performance and slice load balance, but also the future network state and utility, e.g., the queue of packets in the buffer.

Such dynamic characteristics make DRL naturally become a good choice for this problem-solving. Reinforcement learning is a class of machine learning techniques where an agent is trained for decision-making mechanisms by interacting with an environment through action, with the objective of maximizing its cumulative reward. The environment is in a certain state and is changed to the next state by an action from the agent, and the agent receives the corresponding reward for that transition [27]. Therefore, we proposed a well-designed DDPG-Agent to solve the problem. We denote the state, action, state change, and return of the system as the MDP architecture. Given the system state $s_t = s$ and the action $a_t = a$ at time $t$, the next state changes to $s_{t+1} = s'$ with transition probabilities $p_{ss'}^a$ as follows,

$$p_{ss'}^a = P\{s_{t+1} = s' | s_t = s, a_t = a\}, \tag{13}$$

where $p_{ss'}^a$ is influenced by the uncertainty of the previous state, the UEs' mobility, the traffic demands, and the performed action, while part of this, called prior knowledge, is unknown.

A well-designed DRL-based solution of DDPG is proposed to solve the problem dynamically. DDPG can be used to maximize the throughput in wireless communication with a large dimensional continuous action space [28]. It is assumed that the DDPG-Agent is deployed at the BS and the BS can perfectly provide the required state of the system. The design of the DDPG-Agent mainly includes the architecture of the neural network, the observation of the system, and the reward function, as follows.

**State:** The current system state is determined by the serving state of the BS. The proposed DDPG-Agent in this paper takes action at the slice level; therefore, the DDPG-Agent needs to collect information about the state at the slice level and generate resource allocation decisions. The state is defined as $s = \mathsf{Q} \cup \mathsf{E} \cup \mathsf{A}$. $\mathsf{Q} = \{q_m^d | m \in \mathcal{M}, d \in \mathcal{D}\}$ represents a tuple of QoS satisfaction rate of each slice. $\mathsf{E} = \{\eta_m^{d,nr} | m \in \mathcal{M}, d \in \mathcal{D}\} \cup \{\eta^{sul}\}$ represents a tuple of the load of each slice, as well as the SUL band. $\mathsf{A} = \{\alpha_m^d | m \in \mathcal{M}, d \in \mathcal{D}\}$ represents a tuple of the weight of each slice.

**Action:** The action is defined as a tuple $a = \{\delta, \lambda, \mathbf{w}\}$. The well-designed architecture of the actor network and the critic network are shown in Figure 3. $N_{state}/N_{action}/N_{experience}$ represents the dimension of the state/action/experience. The action mapping for taking the output of the actor network as the input of the BS includes action exploring with $\epsilon$-greedy and action clipping. To avoid severe performance degradation due to some extreme actions, we clip the actions to within reasonable limits. Since the bandwidth of SUL is usually much less than that of NR, the constraint on $\delta$ is equivalent to 0.5 to 0.95 of the coverage radius of the BS. The TDD pattern is limited to between 0.2 and 0.8 to account for varying traffic demands. Furthermore, each slice occupies no less than 10% of the total bandwidth $W^{nr}$.

**Reward:** The reward function is the same as (11).

$\pi(s_t | \theta^\pi)$ and $\mu(s_t, a_t | \theta^\mu)$ are used to parameterize the actor network and the critic network, respectively. The corresponding policy is denoted as $\pi$. $Q^\pi(s_t, a_t)$ denotes the value function. The critic network is trained by minimizing the mean-squared Bellman error as follows,

$$L(\theta^\mu) = \frac{1}{B} \times \sum_t (y_t - Q(s_t, a_t | \theta^\pi))^2, \tag{14}$$

where $B$ is the batch size, and the target value $y_t$ is expressed as,

$$y_t = r(s_t, a_t) + \gamma \times \max_{a_{t+1}} Q\left(s_{t+1}, \pi\left(s_{t+1} | \theta^{\pi'}\right) | \theta^{\mu'}\right), \tag{15}$$

where $\theta^{\mu'}$ is the parameter of the target network. The actor network is trained by maximizing the policy objective function $J$, which is described as:

$$\nabla_{\theta^\pi} \approx \frac{1}{B} \times \sum_t \nabla_a Q(s, a | \theta^\mu)|_{s=s_t, a=\theta(s_t)} \nabla_{\theta^\pi} \pi(s | \theta^\pi)|_{s=s_t}. \tag{16}$$

The entire algorithm of DDPG-Agent for the proposed QGJRA-SUL is shown in Algorithm 2.

---

**Algorithm 2** DDPG-Agent for QGJRA-SUL

---

**Input:** The State $s_t$ of the BS includes the QoS satisfaction rate Q, the load E, and the weight A of all slices.

**Output:** An action $a_t$ includes the SUL admission $\delta$, the TDD pattern $\lambda$, and the band slicing scheme **w**.

1: Initialize the Actor Network and the Critic Network; Initialize an empty buffer; Set the number of episode $N_{episode} = 1000$ and the number of step in one episode $N_{step} = 10$; The batch-size $B$ is set to 64;

2: **for** $index_{episode} = 0 \rightarrow N_{episode} - 1$ **do**

3:     state $s_{t-1} \leftarrow$ Reset the environment;

4:     **for** $index_{step} = 0 \rightarrow N_{step} - 1$ **do**

5:         DDPG-Agent $\leftarrow$ the state $s_{t-1}$;

6:         The initial action $a'_t \leftarrow$ DDPG-Agent;

7:         $a''_t \leftarrow$ Exploring the action space with $\epsilon$-greedy $\leftarrow a'_t$;

8:         $a_t \leftarrow$ Action clipping $\leftarrow a''_t$;

9:         The BS of the environment operates for a period $T$;

10:         $s_t, r_t \leftarrow$ environment;

11:         $s_t, a_t, s_{t-1}, r_t \rightarrow$ reply buffer;

12:         $s_t = s_{t-1}$

13:         **if** The number of experiences in the reply buffer< batch-size $B$ **then**

14:             **Pass**;

15:         **else**

16:             Update the DDPG-Agent [29];

17:         **end if**

18:     **end for**

19: **end for**

---

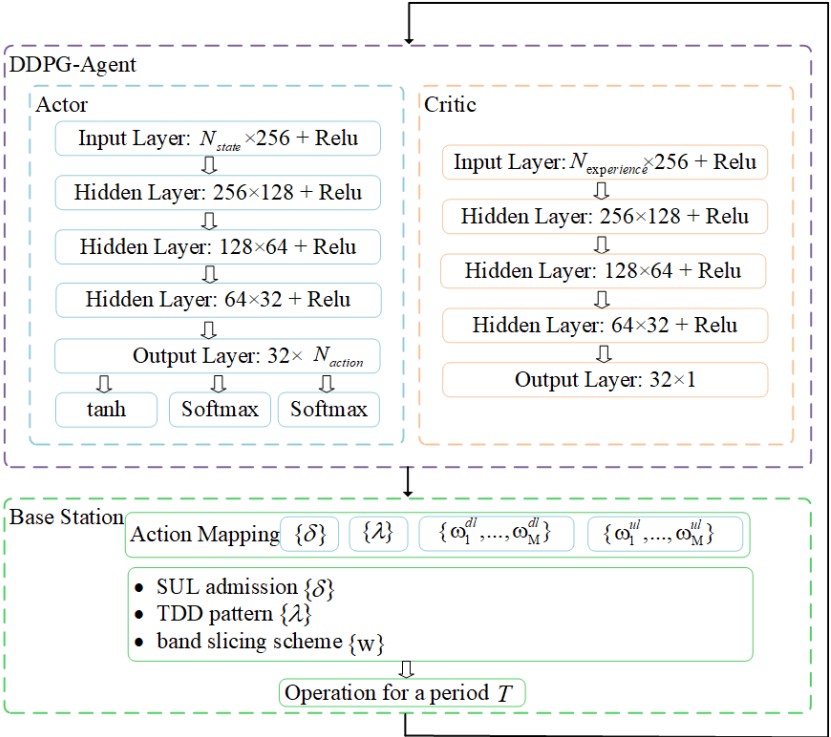

**Figure 3.** The DDPG-Agent is trained for decision-making mechanisms by interacting with the environment. The architecture of the neural network for Actor and Critic for QGJRA-SUL is well designed.

## 5. Numerical Results

### 5.1. Simulation Setting

We perform a Python-based system-level simulation, and the parameters are summarized in Table 2. In this work, three kinds of slices with different downlink and uplink traffic ratios are considered, and the slice parameters are summarized in Table 3. Each slice uses a round-robin algorithm for resource scheduling. In each episode of 10 steps, the number of UEs in each slice is set to increase linearly from 30 to 120 to simulate the load increase. More specifically, 10 UEs are randomly assigned to three slices in each step to simulate a random increase in the number of UEs. Within each episode, the weights of all slices remain the same. In different episodes, the weights of six parts (three slices with downlink and uplink) are increased in turn, while the other five parts share the remaining weights equally. For example, the weight of S1DL is set to 0.35, and the weights of the other five parts are set to 0.13. The size of each packet is set to 12 kbits in the simulation, as the commonly used Maximum Transmission Unit (MTU) is usually set to 1500, that is, 1500 bytes (12 kbits). The arrival of packets satisfies the Poisson distribution. It is assumed that all UEs have the same traffic model. The throughput requirement for each UE is set to be $1.05 \times \bar{R}^d_m$ instead of a common-used full buffer. To ensure that all UEs experience both good channel quality and bad channel quality, UEs are set to move along a straight line. Once a UE reaches the boundary of the given area, it moves in a random direction that does not go outside the boundary. The path loss model of an urban macro scenario is adopted, referring to [2]. The training computer has an Intel core i7-13700K CPU and 32 GB of memory.

**Table 2.** The parameters of the simulation environment.

| Parameters | Value | Parameters | Value |
|:---:|:---:|:---:|:---:|
| $\tau$ | 2 ms | $W^{nr}$ | 100 M (250 RB) |
| $\mu$ | 1 | $W^{sul}$ | 10 M (25 RB) |
| $N^{dL} + N^{uL}$ | 56 | $P^{dL}$ | 43 dBm [1] |
| $T$ | 100 ms | $P^{uL}$ | 23 dBm |
| NR frequency | 3.5 GHz | $N_0$ | −174 dbm/Hz |
| SUL frequency | 1.8 GHz | BS radius | 350 m |

[1] Equal power allocation is used for RAN. The power setting on RB with 20 MHz bandwidth [7] is extended to 100 MHz in this paper.

**Table 3.** Slice Parameters.

|  | S1DL | S1UL | S2DL | S2UL | S3DL | S3UL |
|:---:|:---:|:---:|:---:|:---:|:---:|:---:|
| Packet Size (bits) | 12 k | 12 k | 12 k | 12 k | 12 k | 12 k |
| QoS (Mbps) | 1 | 1 | 2 | 1 | 1 | 2 |

We compare the proposed QGJRA-SUL framework with several baselines, which are listed as follows. Note that the NR bandwidth of all baselines without SUL is set to be 100 MHz + 10 MHz for fairness. In other words, the proposed framework contains 100 MHz for NR and 10 MHz for SUL. Other frameworks labeled as "-nonSUL" contains 100 MHz for NR, and another 10 MHz for NR.

- QGJRA-SUL, $\beta = 0$: The weight value $\beta$ in the penalty term $(-\beta \cdot \varsigma)$ in the reward function (11) is set to 0, which means that the load balancing mechanism is removed.
- QGJRA-nonSUL: QGJRA without the SUL band is tested for comparison.
- DNSIE-nonSUL [30]: A heuristic algorithm based on a fixed threshold realizes resource sharing among different slices.
- Static-NR-nonSUL: The whole QGJRA framework is static with a preset action.
- Optimal: We try almost every possible action to derive the optimal result for QGJRA-SUL.

### 5.2. The Convergence of the DDPG-Agent

To test the influence of coefficient $\beta$ on the convergence speed and the final performance of the DDPG-Agent, $\beta$ was set to 0, 0.5, 1, 2, and 3, respectively. The convergence process of DDPG-Agent under the different values of $\beta$ is shown in Figure 4. The DDPG-Agent converges quickly in the first 500 episodes by exploring the action space. Then, the DDPG-Agent gradually approaches the optimal policy in the following training. Agents with different $\beta$ show similar convergence speeds. Figure 4a shows that the larger the $\beta$, the smaller the corresponding reward value tends to be. To compare the influence of different $\beta$ on the performance of QoS guaranteed, we plotted the value of the reward with the penalty item removed in Figure 4b, that is, the BS-level QoS satisfaction rate. As can be seen, agents with $\beta \neq 0$ show better performances than with $\beta = 0$. Increasing the $\beta$ performance has an obvious marginal effect when $\beta$ is larger than 1. If the reward only has a load balancing mechanism, which is marked as $\beta = 10, \alpha = 0$, the DDPG-Agent will be unable to judge how to achieve a better QoS satisfaction rate after a full load. In the following experiment, $\beta$ is set to 3.

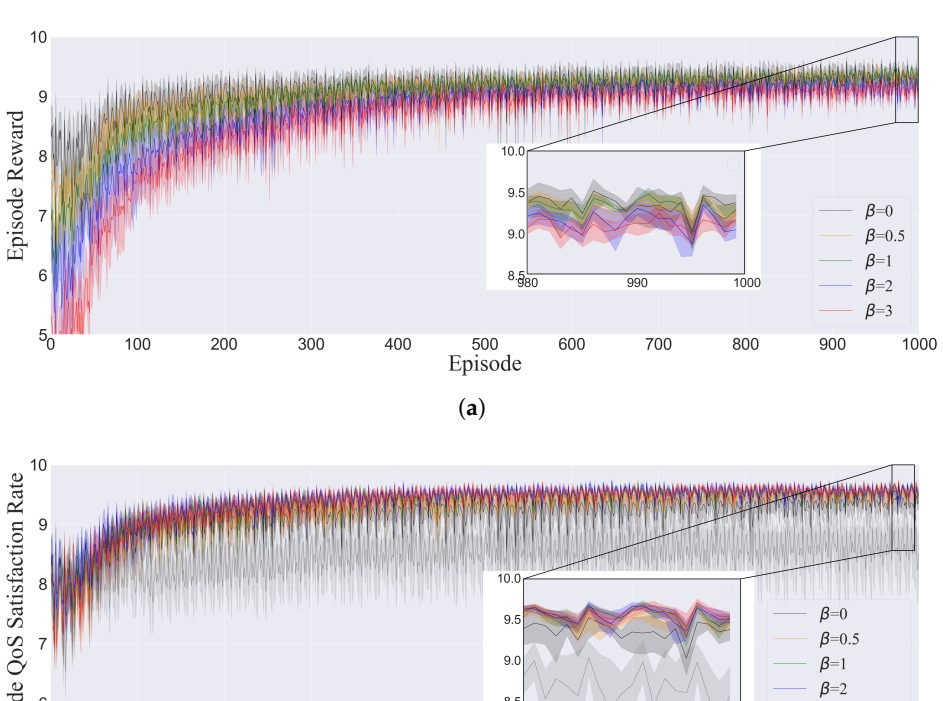

**Figure 4.** The convergence process of DDPG-Agents with different $\beta$. (**a**) The episode reward with different $\beta$ in the convergence process. (**b**) The episode QoS satisfaction rate with different $\beta$ in the convergence process.

### 5.3. Performance Comparison Guarentee of QoS

The BS-level QoS satisfaction rate $q$ is used as a comparison metric because the QoS guarantee is the primary objective of the proposed framework, and the load balancing mechanism is to serve the QoS guarantee. Otherwise, the value of QGJRA-SUL, $\beta = 0$ has an unreasonable advantage over the reward because its reward function removes the penalty item. Figure 5 shows the average QoS satisfaction rate of 100 episodes in the testing. The BS is at full load of about 70 UEs. Compared with other baseline methods, the performance of the proposed QGJRA-SUL is the closest to that of the Optimal. Figure 5 shows an obvious performance gap between QGJRA-SUL and QGJRA-SUL, $\beta = 0$ near the

full load. This gap indicates that the load balancing mechanism can help the DDPG-Agent to make better resource reservation for each slice when the slice is close to the full load. QGJRA-SUL, $\beta = 0$ suffers performance degradation, because it has no load balancing mechanism to deal with more complex traffic fluctuations. This means that considering the load balancing between slices is helpful to improve the QoS satisfaction rate under traffic fluctuations. The edge UEs of QGJRA-nonSUL cannot use the SUL band with less path loss, so its performance is mediocre. In addition, QGJRA-SUL can accommodate about 15% more UEs with the same QoS satisfaction rate as that of QGJRA-nonSUL at a load of 60–70 UEs. The framework with SUL achieves a greater QoS satisfaction rate compared to the framework without SUL because of the lower path loss of the SUL band. DNSIE-nonSUL can share resources among multiple slices based on different load thresholds before overload. However, the granularity of this resource adjustment is fixed, which cannot well match the dynamic traffic requirement.

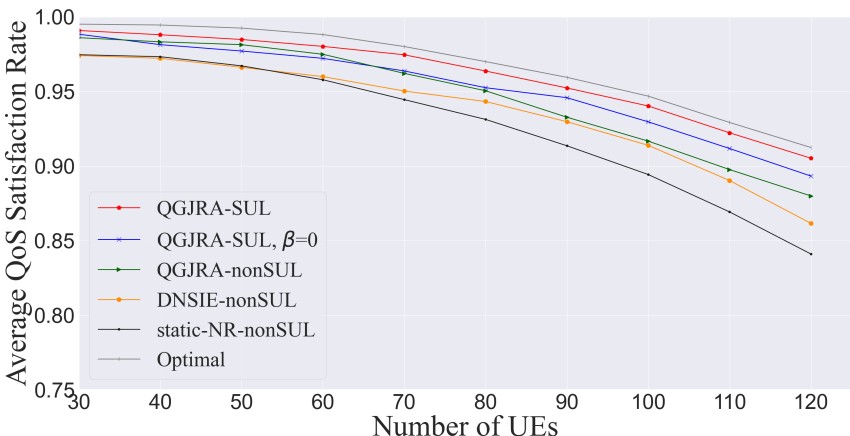

**Figure 5.** The average QoS satisfaction rates of all slices versus the number of UEs, increasing from 30 UEs to 120 UEs.

### 5.4. The Performance of Weight Adaption

To test the sensitivity of the DDPG-Agent to the weight of each slice, we set the number of UEs to 45 per slice to simulate the overload of BS. The setting of $\alpha_m^d$ is shown in Figure 6b. The corresponding slice-level QoS satisfaction rate of each slice is shown in Figure 6a. The DDPG-Agent tends to allocate more resources to the slices with larger weights for a higher QoS satisfaction rate by observing the weights as part of the state, i.e., S2UL has the largest weight during steps 40–50, and it has a higher QoS satisfaction rate than any other time. If a slice with the same weight as the others has a very low value in resource use, it may be sacrificed with the same weight as the others, i.e., S3UL from step 0 to step 50.

Figure 7a shows the percentage of resources allocated to different slices obtained via different slices versus different schemes, and the corresponding slice-level QoS satisfaction rates versus different schemes are shown in Figure 7b. S2DL and S3UL in Table 3 have higher throughput requirements, so the DDPG-Agent allocates more wireless resources to these two parts, which is shown in Figure 7a. Furthermore, the DDPG-Agent allocates more wireless resources to S1UL than to S1DL under the same throughput requirements, because the uplink transmission power is lower than the downlink transmission power. Although the heuristic-based DNSIE-nonSUL scheme has similar resource allocation results to that of the DRL-based QGJRA-nonSUL scheme, the DRL-based solution can achieve a better performance in QoS, as guaranteed through dynamic and detailed parameter adjustment. DDPG-driven QGJRA-nonSUL is able to achieve the performance gains of more slices, with a small performance loss of a few slices. The performance of the static-NR-nonSUL scheme is completely determined by preset actions, and it cannot adapt to complex traffic fluctuations, resulting in the worst performance of the BS-level QoS satisfaction rate.

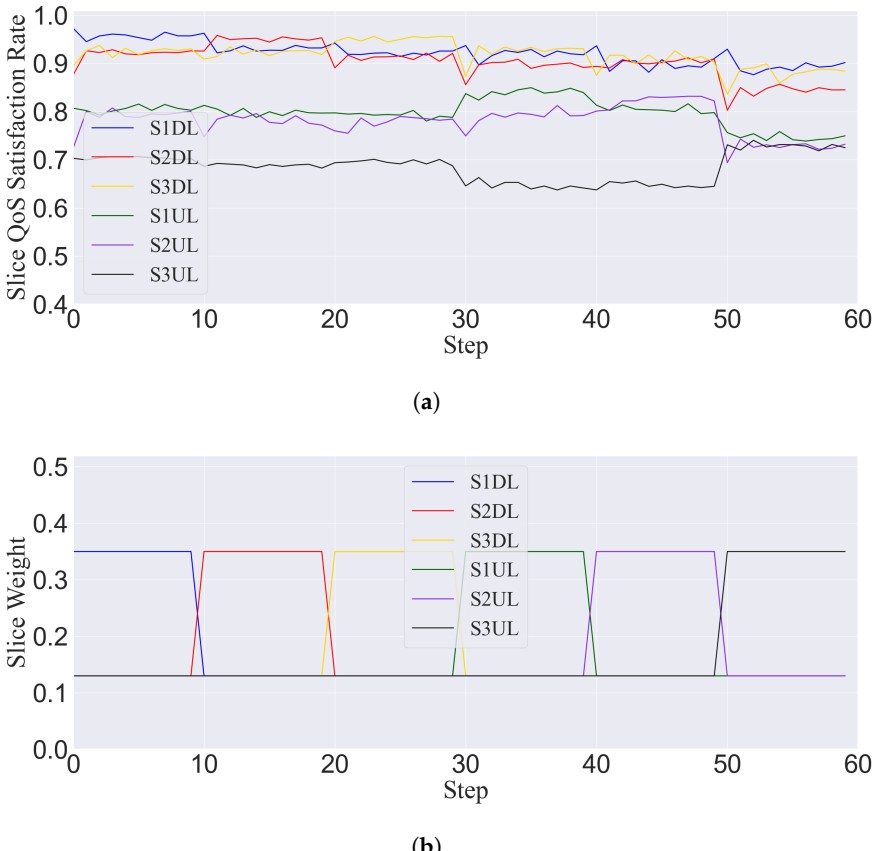

**Figure 6.** The adaptation of QoS for different weights. The weights of each slice in turn become the maximum value, and the others remain the same. (**a**) The slice-level QoS satisfaction rate of each slice. (**b**) The weights of each slice.

*5.5. Performance Comparison of Load Balancing*

Finally, we have tested the effect of $\beta$ on the load-balancing mechanism. The performance of slice load balancing is tested in terms of the average variance of the slice load, where QGJRA-SUL with the variance of the slice load $(-\beta \cdot \varsigma)$ as a penalty term and QGJRA-SUL without a penalty term are compared in 200, 400, 600, 800, and 1000 episodes, respectively. The number of episodes where the DDPG-Agent is trained is represented by $N_{episode}$. As shown in Figure 8, all curves show first rise and then fall, corresponding to the BS load from light loads to medium loads, and then to full loads. The loads of slices are more balanced at low and full loads because resources are not being used much, or are all being used. The main factors affecting the QoS guarantee of slices are resource allocation policy near the full load. The performance in the load balancing of QGJRA-SUL with the penalty term outperforms that of QGJRA-SUL without the penalty term. obviously. The average variance of slice load $\varsigma$ is 0.0465 for $\beta = 0$ and 0.0125 for $\beta = 3$ at step 4, respectively. The DDPG-Agent with $\beta = 3$ achieved a 73% increase in the performance of load balancing than that with $\beta = 0$ near the full load. The reward function with the penalty term can lead the DDPG-Agent to realize load balancing well. In addition, in the training process, QGJRA-SUL with the penalty term can make DDPG-Agent learn the QoS satisfaction and load balancing of slices simultaneously, which can avoid the QoS satisfaction degradation caused by some slices being overloaded and some slices being underloaded.

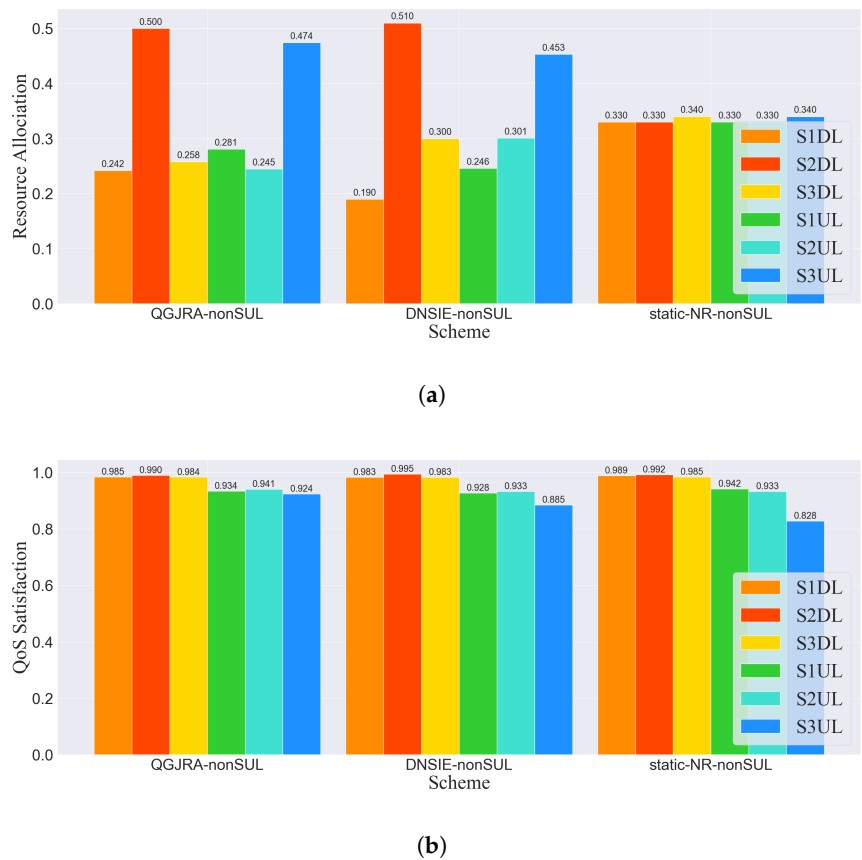

(**a**)

(**b**)

**Figure 7.** Average QoS satisfaction of each slice in 10 episodes at an NR bandwidth of 110 MHz. (**a**) The band slicing schemes of different methods. (**b**) The QoS satisfaction rate of each slice under different methods.

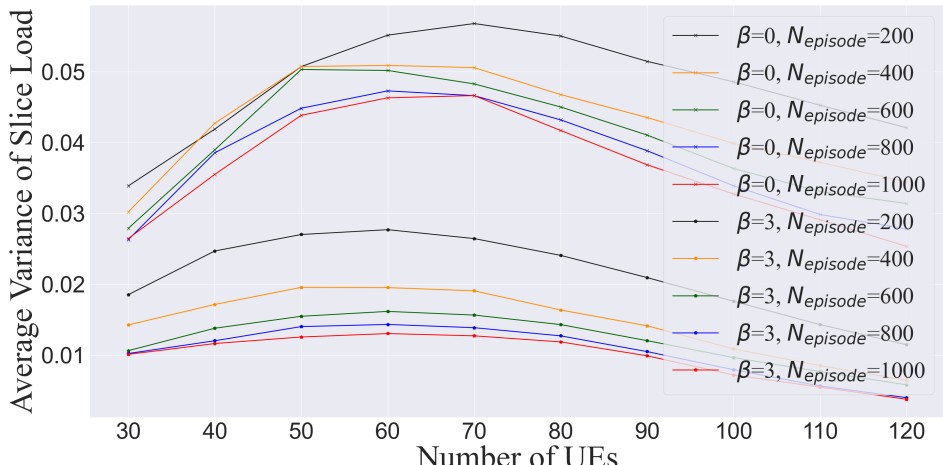

**Figure 8.** The variance of load versus the number of UEs under a different number of training episodes.

### 5.6. Complexity Analysis

In this subsection, we provide some computational complexity analysis of the DDPG used in this paper. The DNN network of this paper employs the full-connection networks, the computational complexity of each training step, including the part of the actor network and the part of the critic network. In actual operation, only the trained actor network is needed. The computational complexity of each step of the actor network is

$O(\sum_{l=1}^{L} K_{l-1}K_l)$, where $K_l$ represents the neural size of the *l*th layer ($1 \le l < L$) among *L* layers. The architecture of the neural network is shown in Figure 5.

## 6. Conclusions

In this paper, we propose a novel QGJRA-SUL framework for 5G NR with SUL, where three parameters of SUL admission, TDD pattern, and the band slicing scheme are jointly optimized. We also consider the load balancing between different slices in our framework, which further improves the network QoS satisfaction rate. The framework is driven by a well-designed DDPG-Agent. By combining the activation functions tanh and softmax, the DDPG-Agent can optimize all three parameters at the same time. Under the original problem of maximizing the QoS satisfaction rate, we introduce the load unbalance degree of slices into the reward function as a penalty term. In the simulation, we explore the features of the framework, including QoS guaranteed, weight adaptability, and load balancing. The simulation results show that the proposed framework can guarantee the QoS satisfaction rate under rapidly fluctuating traffic demands while maintaining the load balance of slices well. The proposed QGJRA-SUL can accommodate 15% more UEs with the same QoS satisfaction rate than that of a traditional single-band solution without SUL, and achieve a 73% increase in the performance of load balancing than that without a load balancing mechanism near the full load.

**Author Contributions:** Conceptualization, Y.S.; methodology, Y.S.; software, Y.H.; validation, S.Z.; writing—original draft preparation, Y.H.; writing—review and editing, Y.H., T.Y., X.C., and S.Z. All authors have read and agreed to the published version of the manuscript. All authors have read and agreed to the published version of the manuscript.

**Funding:** This work was supported by the Innovation Program of Shanghai Municipal Science and Technology Commission under Grant 22511100604 and 20JC1416400; the National Key Research and Development Program of China under Grants 2022YFB2902005, 2022YFB2902304, and 2022YFB2902002; the National Natural Science Foundation of China (NSFC) under Grants 62071284 and 61904101; and the Key-Area Research and Development Program of Guangdong Province under Grants 2020B0101130012.

**Data Availability Statement:** Data sharing not applicable.

**Conflicts of Interest:** The authors declare no conflicts of interest.

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
