# Peer review of "A Novel QoS Guaranteed Joint Resource Allocation Framework for 5G NR with Supplementary Uplink Transmission"

_electronics, doi:10.3390/electronics12071563_

Round 1
Reviewer 1 Report
1. The authors proposed entitled “A Novel QoS Guaranteed Joint Resource Allocation Framework for 5G NR with Supplementary Uplink Transmission”. Authors should add the algorithms of the system model in Chapter 3.
2. First, the content of the background and goal of the proposal is insufficient, and the part of the simulation result is very insufficient. In particular, performance results with other methods will improve the quality of the proposed paper.
3. The characteristics, advantages, and relationship of the proposed Supplementary Uplink (SUL) in 5G NR with other systems should be presented in more detail.
4. Improve the spectrum of simulations a bit more.

Reviewer 2 Report
This article proposed a QGJRA-SUL framework based on deep reinforcement learning for 5G NR scenarios with supplemental uplink, which jointly optimizes SUL access, TDD mode, and bandwidth slicing schemes. In all, this manuscript seems to be technically solid and easy to follow. Some minor revisions need to be considered as follows:
1. The abstract needs to provide more details for the proposed resource allocation framework, such as the design details.
2. Authors are suggested to make great efforts to update references and introduce more details about the state-of-the-art works. Some recent works about resource allocation or deep reinforcement learning are suggested to be introduced such as [1] Impacts of sensing energy and data availability on throughput of energy harvesting cognitive radio networks, IEEE Transactions on Vehicular Technology, 2023. [2] DDPG-based joint time and energy management in ambient backscatter-assisted hybrid underlay CRNs, IEEE Transactions on Communications, 2023. [3] Throughput maximization of wireless-powered communication network with mobile access points, IEEE Transactions on Wireless Communications, 2023.
3. The python-based system-level simulation should be introduced in detail.
4. The quality of numerical result figures can be improved a lot, especially Fig. 7.
5. The format of the manuscript looks quite unappealing, and the presentation of mathematical symbols appears to be informal. It is recommended to use a standard LaTeX template to re-edit the manuscript.
Reviewer 3 Report
Dear Authors,
Your paper entitled "A Novel QoS Guaranteed Joint Resource Allocation Framework for 5G NR with Supplementary Uplink Transmission" shows a new resource management method in order to optimize the quality of service. In the manuscript, you also provided a quantitative analysis with the help of numerical simulation in order to investigate the proposed communication model (MDP).
Overall, the paper is highly detailed with sufficient description in the introductory chapter, system model chapter, problem formulation chapter, problem transformation chapter, Numerical Results chapter. The
My concerns regarding your paper are presented in the rows below:
1. First, I am used to your form of referencing the work of others. I present some of references in the introductory chapter:
studied in [] line 25, association in [] line 27, in [] and [] line 33, discussed in [] line 41. I do not recommend this form of appreciation, referencing others work. I recommend reading a recently published work in electronics journal ( https://www.mdpi.com/2079-9292/12/5/1218 ) to see the differences of references with your manuscript.
I recommend that you will remodel/rephrase the manuscript in light of these differences.
2. Second, the manuscript misses the limits of the study and assumptions. A paragraph in: the system model, problem formulation, problem transformation, and Numerical results chapters would help increase the readability of the paper.
3. Third, another concern is regarding the setup used for simulations. No description was provided in this sense. For reproducibility of the results please provide the setup description and environment description used in your computations in your simulations.
4. What is the number of clients(numbers of user's equipment) that are estimated in the given investigated area of 350 x 350 m in the section numerical results? You used the term episode/episodes and Wnr (The bandwidth of the NR band) in table 3 with the value of 100M, Wsul 10M, however it is not clear the connection presented in 5.3 with the parameters of the simulations and slice parameters. Please provide an in-depth description to accommodate the support for QoS satisfaction results.
5. Conclusion section is too narrow, thus your results are not properly highlighted. Please take the time and provide your best additions to the research presented in the manuscript.
I am aware that the new mobile generation 5G is capable of high data rates and also that some of the recipients of this communication are not equally serviced.
I really this that your research is novel through the proposed model and algorithm and can help accommodate higher QoS levels.
Good luck with your research!
Best regards
Round 2
Reviewer 1 Report
Comments and Instructions
The authors carefully followed the comments and instructions of a reviewer.
However, the authors wish to supplement the following comments.
3. The characteristics, advantages, and relationship of the proposed Supplementary Uplink (SUL) in 5G NR with other systems should be presented in more detail. (For Example. Table)

Reviewer 3 Report
Dear Authors,
I really appreciate that you have taken into account my observations.
As a matter of fact, the manuscript quality has increased significantly.
Here are some of new additions that:
- The abstract is more precise on the subject: Deep reinforcement learning engine
- Introduction has new additions on why this research is important, towards the subject of maximizing QoS satisfaction rate through network slicing and Supplementary Uplink (SUL).
- Also, additional explanations and better captions in sections 2, 3, 4 increase the visibility of the figures and algorithms
- Section 5 has a better description of what system was used to process the whole algorithm.
- Figure 6 b on slice weight distribution is a nice addition
- Figure 7 also has better captioning
- The conclusion section additions also highlight the use of DDPG-Agent to drive the algorithm.
It would be great in the near future to confirm your findings with real experiments of 5G user devices like in Figure 7 and 8.
Good luck with your research!
